# *Chlorella pyrenoidosa* Polysaccharides as a Prebiotic to Modulate Gut Microbiota: Physicochemical Properties and Fermentation Characteristics In Vitro

**DOI:** 10.3390/foods11050725

**Published:** 2022-03-01

**Authors:** Kunling Lv, Qingxia Yuan, Hong Li, Tingting Li, Haiqiong Ma, Chenghai Gao, Siyuan Zhang, Yonghong Liu, Longyan Zhao

**Affiliations:** 1College of Light Industry and Food Engineering, Guangxi University, Nanning 530004, China; kunlinglv@foxmail.com; 2Institute of Marine Drugs, Guangxi University of Chinese Medicine, Nanning 530200, China; qingxiayuan@163.com (Q.Y.); hongli12212022@163.com (H.L.); li15578909861@126.com (T.L.); MHQ18878839254@163.com (H.M.); gaochh@gxtcmu.edu.cn (C.G.); yonghongliu@scsio.ac.cn (Y.L.)

**Keywords:** *Chlorella pyrenoidosa*, polysaccharides, fermentation, gut microbiota, prebiotic

## Abstract

This study was conducted to investigate the prebiotic potential of *Chlorella pyrenoidosa* polysaccharides to provide useful information for developing *C. pyrenoidosa* as a green healthy food. *C. pyrenoidosa* polysaccharides were prepared and their physicochemical characteristics were determined. The digestibility and fermentation characteristics of *C. pyrenoidosa* polysaccharides were evaluated using in vitro models. The results revealed that *C. pyrenoidosa* polysaccharides were composed of five non-starch polysaccharide fractions with monosaccharide compositions of Man, Rib, Rha, GlcA, Glc, Gal, Xyl and Ara. *C. pyrenoidosa* polysaccharides could not be degraded under saliva and the gastrointestinal conditions. However, the molecular weight and contents of residual carbohydrates and reducing sugars of *C. pyrenoidosa* polysaccharides were significantly reduced after fecal fermentation at a moderate speed. Notably, *C. pyrenoidosa* polysaccharides could remarkably modulate gut microbiota, including the promotion of beneficial bacteria, inhibition of growth of harmful bacteria, and reduction of the ratio of Firmicutes to Bacteroidetes. Intriguingly, *C. pyrenoidosa* polysaccharides can promote growth of *Parabacteroides distasonis* and increase short-chain fatty acid contents, thereby probably contributing to the promotion of intestinal health and prevention of diseases. Thus, these results suggested that *C. pyrenoidosa* polysaccharides had prebiotic functions with different fermentation characteristics compared with conventional prebiotics such as fructooligosaccharide, and they may be a new prebiotic for improving human health.

## 1. Introduction

Microalgae have enormous potential to be a food source and a new crop to resolve the world food problem in the 21st century [1]. *Chlorella*, one of the most widely cultivated species of microalgae, has also attracted considerable attention. It possesses many attractive features, as most microalgae do, such as easy cultivation, a fast growth rate, resistance to adverse growth conditions, rich nutrient value, and a high content of various biologically active substances. There is considerable evidence that *Chlorella* has a variety of biological activities, such as anti-tumor, anti-oxidant, immunoregulatory, and anti-diabetic activities [2,3]. Therefore, *Chlorella* is listed as a “green healthy food” by the Food and Agriculture Organization of the United Nations (FAO), and many products are continually being developed by pharmaceutical companies and food processing factories. However, there are some improvements that will need to be made before *Chlorella* can become a regular food source. Especially, the metabolism and health effects of some macromolecules, such as polysaccharides and proteins from the *Chlorella*, following oral administration should be deeply evaluated to further supplement and improve the human diet.

*Chlorella* contains a high number of polysaccharides as one of its most abundant active ingredients. These compounds from various *Chlorella* species with various structures and biological activities have been reported in recent decades, which are comprehensively summarized in our previous review [3]. *Chlorella* polysaccharides can be divided into neutral, acidic, and amino polysaccharides, with various types of monosaccharides and glycosidic linkages. Notably, as shown in the review, many studies have shown that *Chlorella* polysaccharides have a wide range of bioactivities after oral administration, such as immunomodulatory [4,5] and hypolipidemic activity [6]. However, it has been widely recognized that most polysaccharides with a high molecular weight have difficulty crossing the epithelial cells of the gastrointestinal (GI) tract into the systemic circulation [7,8], and are mainly excreted in the feces after oral administration [9]. Hence, how these macromolecule polysaccharides from *Chlorella* change and exert their activities after oral administration still needs to be investigated.

As reported, some macromolecular polysaccharides from natural sources can be digested by saliva or gastrointestinal fluid, and some cannot [10]. However, studies on the digestibility of polysaccharides even from same species offer conflicting conclusions. For example, it was shown that polysaccharides from sea cucumber could be digested by simulated gastric and intestinal fluid [11,12], but there was a study indicated that these polysaccharides cannot be degraded by gastrointestinal digestion [13]. Thus, the changes of polysaccharides under saliva and gastrointestinal conditions need to be further investigated. Increasing studies have indicated that polysaccharides from natural sources may be degraded and utilized by gut microbiota. Previous studies also showed that *Chlorella* possessed prebiotic effects. The addition of *Chlorella* could not only promote the growth and viability of probiotic bacteria, but also inhibit intestinal pathogens [14]. The main chemical components from *Chlorella* possessing prebiotic effects, however, remain unknown. According to previous studies, we hypothesized that *Chlorella* polysaccharides may be a prebiotic able to exert its activities in the gastrointestinal tract.

To date, no information is available on the digestibility, fermentation characteristics, and effects on the gut microbiota of *Chlorella* polysaccharides. The number of gut microbiota is more than 100 trillion, and its genes exceed 3 million, which may encode abundant glycosidases to hydrolyze polysaccharides. Some low-molecular-weight oligosaccharides derived from polysaccharides may be absorbed by the human body and then perform their biological activities. In addition, metabolites such as short chain fatty acids (SCFAs) produced by gut microbiota after fermenting polysaccharides and oligosaccharides may also have critical functions within the human body.

Generally, some plant tissues contain a lot of starch polysaccharides. Studies on the polysaccharides from plants containing starch may be difficult to elucidate their digestibility and fermentation characteristics. Thus, in this study, non-starch *Chlorella* polysaccharides were prepared and characterized. The digestion characteristics, including changes of molecular weight, total carbohydrates, reducing sugars and free monosaccharides of the *Chlorella* polysaccharides in saliva and gastrointestinal medium were evaluated by introducing an in vitro digestion model. Furthermore, the detailed variations of molecular weight and carbohydrate consumption of the *Chlorella* polysaccharides, and their pH value, microbial composition, and SCFAs production were explored using human fecal inoculum in vitro. These results will facilitate the application of *Chlorella* polysaccharides in functional foods for their prebiotic effects.

## 2. Materials and Methods

### 2.1. Materials and Reagents

The powder form of *Chlorella pyrenoidosa* was obtained from Shanghai Guangyu Biological Technology Co., Ltd. (Shanghai, China). Pepsin, α-amylase, lipase, pancreatin, 3-Methyl-1-phenyl-2-pyrazolin-5-one (PMP), and the standard monosaccharides including rhamnose (Rha), glucuronic acid (GlcA), galacturonic acid (GalA), glucose (Glc), and galactose (Gal) were purchased from Sigma-Aldrich (St. Louis, MO, USA). Acetic, propionic, butyric, valeric and 2-ethylbutyric acid with chromatographic purity were purchased from Aladdin Biochemical Technology Co., Ltd. (Shanghai, China). Fructooligosaccharide (FOS) was obtained from Macklin Inc. (Shanghai, China). All other chemicals and solvents were of analytical grade and obtained commercially.

### 2.2. Polysaccharide Extraction

The *C. pyrenoidosa* powder was mixed with distilled water at a ratio of 1:20 (*w*/*v*) and extracted at 90 °C for 3 h. After centrifugation, the precipitation was extracted twice more under the same conditions. The supernatant was combined, treated with amylase, and then precipitated by 95% ethanol in sequence. After centrifugation, the precipitate was re-dissolved, and its proteins were removed via the Sevag method. The obtained solution was dialyzed against distilled water (molecular weight cut-off 3.5 kDa) for 3 days. Finally, the dialysate was concentrated and lyophilized to obtain the CPP.

### 2.3. Physicochemical Analysis

The total carbohydrate content was determined by the phenol-sulfuric method as described by Dubois et al. [15] using Glc, Gal, and Man as standards. The protein content was determined using a modified method from Bradford et al. [16]. A hydroxydiphenyl assay established by Blumenkrantz and Asboe-Hansen (1973) was used to analyze the uronic acid content [17]. The concentration of the reducing sugars was determined using a dinitrosalicylic acid (DNS) method [18]. The sulfate content of CPP was determined by a classical turbidimetric method [19]. Elemental analysis of CPP was performed on a Vario EL III elemental analyzer (Elementar, Langenselbold, Germany) to analyze the carbon, hydrogen, nitrogen, sulfur, and oxygen contents.

The molecular weight distribution of samples was analyzed by an LC-2030C 3D HPLC (Shimadzu Corp., Kyoto, Japan) with a refractive index detector (RID) using a Shodex OHpak SB-804 HQ column (7 µm, 8 × 300 mm). The NaCl solution (0.1 M) was used as eluent at a flow rate of 0.5 mL/min, and the column temperature was maintained at 35 °C. The pullulan standards (p1–5) were used to estimate the molecular weight of CPP. The molecular weights of p1–5 were 9.6, 21.1, 47.1, 107.0, and 344.0 kDa, respectively.

The monosaccharide composition was determined based on our previous report [20], with minor modifications. The samples were derivatized by PMP at 70 °C for 100 min. The derivatives were identified using HPLC equipped with a DAD detector. The separation was performed on an Agilent Zorbax Eclipse Plus C18 column (4.6 × 250 mm, 5 μm) (Agilent Technologies Co., Ltd., Santa Clara, CA, USA) using 20 mM ammonium acetate solution and acetonitrile (83:17, *v*/*v*) as the mobile phase at a flow rate of 1.0 mL/min. The column temperature was maintained at 30 °C.

The IR spectrum of CPP (2 mg) was determined through KBr pellets by a Nicolet iS50 Fourier transform infrared spectroscopy (FT-IR) spectrometer (Thermo Fisher Scientific, Wilmington, MA, USA) in the range of 4000–400 cm^−1^ at room temperature.

Using a Bruker Advance 600 MHz spectrometer equipped with a ^13^C/^1^H dual probe in FT mode, as described in our previous study [21], the ^1^H NMR spectrum was recorded at 298.1 K. The CPP was dissolved in deuterium oxide (D_2_O, 99.9% D) at 20 mg/mL.

### 2.4. Simulated Saliva Digestion

The in vitro simulated saliva digestion of CPP was carried out according to the previously reported method, with minor modifications [22]. The simulated salivary medium was prepared by dissolving NaCl (0.12 g), KCl (0.15 g), α-amylase (2.0 g), and mucin (1.0 g) in 1.0 L of deionized water, and by adjusting the pH to 7.0 by adding 0.1 M NaOH solution. The CPP solution (8 mg/mL) was mixed with the simulated salivary medium at the same volume, and then the mixture was incubated at 37 °C in a water bath. During incubation, the samples were taken out at 0, 0.5, 1.0, and 2.0 h, subjected to a boiling water bath for 10 min to inactivate the enzyme activity, and used for further analysis.

### 2.5. Simulated Gastrointestinal Digestion

The artificial gastric digestion of CPP was executed according to the method described previously [23]. Briefly, the gastric electrolyte solution (200 mL) containing NaCl (620.0 mg), KCl (220.0 mg), NaHCO_3_ (120.0 mg), and CaCl_2_ (30 mg) was prepared, and its pH was adjusted to 2.0 by the addition of 0.1 M HCl solution. Then, 50 mg of gastric lipase, 47.2 mg of gastric pepsin, and 2 mL of CH_3_COONa (1 M, pH 5) were added to the prepared solution, and the pH was adjusted to 2.0 with the addition of 0.1 M HCl solution to obtain an artificial gastric medium. The CPP solution was mixed with the artificial gastric medium at a ratio of 1:1 (*v*/*v*), and then incubated at 37 °C in a water bath. At 0, 2, 4, and 6 h of the digestion process, samples were taken out and boiled at 100 °C for 10 min.

The artificial small intestinal digestion of CPP was performed according to the method described by Zhou et al. [24], with minor modifications. The small intestinal electrolyte solution was made up of NaCl (5.4 g/L), KCl (0.65 g/L), and CaCl_2_ (0.33 g/L). The pH was adjusted to 7.0 with the addition of 0.1 M NaOH solution. One hundred grams of the bile salt solution (4%, *w*/*w*), 50 g of pancreatin solution (7%, *w*/*w*), and 6.5 mg of trypsin were mixed thoroughly with 50 g of the small intestinal electrolyte solution. The mixture was then neutralized to pH 7.0 to obtain the artificial small intestinal medium. After the artificial gastric digestion, the solution was neutralized to pH 7.0 again, and mixed with the artificial small intestinal medium at a ratio of 10:3 (*v*/*v*). The samples were taken out after digestion for 0, 2, 4, and 6 h, boiled in a water bath for 10 min, and analyzed.

### 2.6. In Vitro Fermentation

The in vitro fermentation of CPP was conducted under anaerobic conditions based on the methods previously reported [25]. The fresh feces were collected from five healthy adults (20–26 years old, two females and three males) who had no history of gastrointestinal diseases and did not receive any treatment with antibiotics within the preceding 3 months. The feces were immediately transferred to an anaerobic tube and mixed with the sterile modified saline (containing 0.5 g/L cysteine-HCl and 9.0 g/L NaCl) to obtain a 10% fecal slurry (*w*/*v*). The fecal slurry was further centrifugated. The supernatant was collected and mixed evenly to obtain the final human fecal inoculum. The basal nutrient medium (1.0 L) was prepared using yeast extract (2.0 g), peptone (2.0 g), NaHCO_3_ (2 g), L-cysteine (0.5 g), bile salts (0.5 g), NaCl (0.1 g), KH_2_PO_4_ (0.04 g), K_2_HPO_4_ (0.04 g), hemin (0.02 g), MgSO_4_ (0.01 g), CaCl_2_ (0.01 g), 2.0 mL of Tween-80, 1.0 mL of resazurin solution (1%, *w*/*v*), and 10 μL of vitamin K. The pH of the mixture solution was adjusted to 7.0, and then autoclaved at 120 °C for 20 min. One hundred milligrams of CPP or FOS in 9 mL of the basal nutrient medium was mixed with 1.0 mL of the fecal inoculum to begin the fermentation process. In this experiment, the CPP or FOS was used as the sole carbon source, and the medium without carbon sources served as a blank control. Subsequently, all the mixtures were incubated in an anaerobic chamber with a thermostat shaker at 37 °C for 48 h. The fermentation products of the samples were collected at 0, 6, 12, 24, and 48 h for further analysis.

### 2.7. Determination of pH

The pH values of the samples before and after fermentation were determined using a pH meter (Mettler-Toledo Instruments Co., Ltd., Shanghai, China).

### 2.8. Analysis of Short-Chain Fatty Acids (SCFAs)

The supernatant of the fermentation was mixed with the equal volume of 0.2 M HCl solution containing 2-ethylbutyric acid as the internal standard. The compositions and levels of SCFAs in the mixtures were analyzed using GC-MS (Agilent Technologies Co., Ltd., Santa Clara, CA, USA) equipped with an Agilent DB FFAP column (30 m × 0.25 mm × 0.25 μm). The flow rate of helium was 1.0 mL/min. The temperature program started at 100 °C, increased by 5 °C/min to the final temperature of 180 °C, and held for 4 min. The temperatures of the injector and ion source were maintained at 250 and 230 °C, respectively.

### 2.9. Analysis of Gut Microbiota

After 48 h of fermentation, the DNA in different groups was extracted by a DNA kit (Omega Biotek, GA, USA) following the manufacturer’s instructions. The DNA quality was detected by a NanoDrop 2000 UV–vis spectrophotometer (Thermo Fisher Scientific, Wilmington, MA, USA) and the 1% agarose gel electrophoresis. The V3–V4 hypervariable regions of the bacteria 16S rRNA were amplified with the primers 341F (5′-CCTACGGGNGGCWGCAG-3′) and 806R (5′-GGACTACHVGGGTATCTAAT-3′) by a PCR system. The PCR products were then purified and quantified, and the sequencing libraries were generated. After that, the library was sequenced on an Illumina navoseq PE250 platform according to the standard protocols by Gene Denovo Biotechnology Co., Ltd. (Guangzhou, China). All the results were based on the sequenced reads and the operational taxonomic units (OTUs).

### 2.10. Statistical Analysis

Each experiment was done in triplicate, and the data were expressed as the mean ± SD. Statistical analyses were performed using one-way analysis of variance (ANOVA) using IBM SPSS statistics version 26.0. *p* values less than 0.05 (*p* < 0.05) were considered to be statistically significant.

## 3. Results and Discussion

### 3.1. Physicochemical Characteristics of CPP

The content of total carbohydrates of the CPP determined using Glc, Gal, and Man as standards was 51.2 ± 6.4%, 65.6 ± 1.3%, and 43.8 ± 0.9%, respectively. The contents of protein, sulfate group, and uronic acid in the CPP were 0.4 ± 0.1%, 4.6 ± 0.2%, and 12.0 ± 0.9%, respectively. The content of sulfur and nitrogen was 1.4% and 2.9%, respectively, indicating the existence of sulfated and amino polysaccharides in the CPP. The high-performance gel permeation chromatography (HPGPC) profile showed that the CPP mainly consisted of five polysaccharide fractions of different molecular weight in a range from 15.3 to 571.8 kDa (Figure 1A). It was composed of Man, Rib, Rha, GlcA, Glc, Gal, Xyl, and Ara with a molar ratio of 1.19:0.55:1.65:1.80:1.13:2.97:0.44:2.27 (Figure 1B). In a report from Chen et al. [26], the polysaccharide from *C. pyrenoidosa* may be also a polysaccharide mixture of different molecular weight and was composed of Glc, Man, Gal, Ara, Xyl, and Rha, with a molar ratio of 25.3:7.2:5.0:3.8:1.0:0.4. As was known, the polysaccharides from *C. pyrenoidosa* contained starch [3]. However, the polysaccharides were not treated with amylase during their extraction process. A lot of starch may exist in their obtained polysaccharides. Therefore, the physicochemical characteristics, such as monosaccharide composition, were obviously different from those of the CPP prepared in our study.

The IR spectrum of the CPP is shown in Figure 1C, and the signals can be assigned according to our previous study [21]. The wide and strong absorption band at about 3421 cm^−1^ could be assigned to the stretching vibration of the OH group. The absorption peak at 2924 cm^−1^ could be ascribed to the C-H stretching vibration of -CH_3_ of Rha. The absorption at 1647 cm^−1^ was due to the asymmetric stretching vibration of the C=O of GlcA. The 1417 cm^−1^ absorption peak was COO- asymmetric stretching vibration within GlcA. The absorption at 1045 cm^−1^ was the stretching vibration of C-O of the sugar ring. In addition, the absorption peaks at 1251 cm^−1^ and 852 cm^−1^ were the stretching vibration of S=O and the bending vibration of C-O-S within sulfate, respectively.

The ^1^H NMR spectrum of CPP is shown in Figure 1D. The signals observed at 1.3–1.4 ppm could be assigned to the methyl protons of Rha. The signals at 2.0–2.3 ppm were due to the methyl protons of -(CO)CH_3_, indicating that some of the monosaccharide residues of the CPP were *O*-acetylated. The level of *O*-acetylation could be calculated as 12.5% using the methyl and anomeric proton peak area ratios in the ^1^H NMR spectrum, which are from the acetyl groups and the monosaccharide residues, respectively. The broad and overlapping signals at the region of 3.3–4.4 ppm could be assigned to the ring protons of the sugar residues. The signals at 5.0–5.5 ppm and 4.5–4.7 ppm can be attributed to the anomeric protons of α and β configurations of sugar residues, respectively. According to the results of the ^1^H NMR analysis, few signals of impurities were observed, indicating that the CPP had a high purity.

### 3.2. Characterization of CPP during Simulated Saliva and Gastrointestinal Digestion

The oral cavity is the first digestive site, and amylase plays an important role in breaking down food. As shown in Appendix A, the retention time and response value of the CPP were almost unchanged after the simulated saliva digestion. The contents of total carbohydrates and reducing sugars were further determined. There were no obvious changes in the contents of total carbohydrates and reducing sugars throughout the saliva digestion period. Furthermore, there were no free monosaccharides produced from the CPP after the saliva digestion. Thus, the results indicated that the simulated saliva could not degrade the CPP.

It has been reported that some polysaccharides could not be affected by salivary amylase, but could be hydrolyzed under the simulated gastrointestinal condition due to the low pH environment and digestive enzymes [27,28,29]. Therefore, in vitro simulated gastric and intestinal digestion was also performed in this study. As shown in Appendix A, the molecular weight of the CPP remained constant even after 6 h of simulated gastrointestinal digestion. During the gastric and intestinal digestion, both total carbohydrate and reducing sugar contents did not change significantly (Appendix A). In addition, no free monosaccharides were observed in the chromatograms of the CPP during gastrointestinal digestion. Based on these results, it was confirmed that the artificial gastric and small intestinal medium had little effect on the glycosidic bond of the CPP. Similar results were observed in other studies on polysaccharides from natural sources [30,31], which may be due to these polysaccharides having an unusual structure and not containing starch. Notably, starch can be easily digested by digestive juice, which influences the experimental results to make a correct conclusion.

### 3.3. Characterization of CPP during In Vitro Fermentation

The polysaccharides utilized by gut microbiota can be reflected by the changes in their molecular weight. Therefore, the molecular weight changes of CPP at different time points during fermentation were determined in this study. As shown in Figure 2A,B, the retention time of the CPP was kept constant, while the peak proportion gradually decreased in the progress of fermentation, indicating that CPP was tardily utilized by the fecal microbiota. The proportion of the total carbohydrates at different time points during the fermentation process of the CPP was determined and is shown in Table 1. The contents of the total carbohydrates in the fermentation juices gradually decreased with the increase in fermentation time. After 48 h of fermentation, the remaining amounts of the CPP were determined to be 44.20 ± 2.65%, suggesting that half of the CPP was utilized by gut microbiota. The CPP contained a high content of reducing ends, and the content of the reducing sugars also decreased, obviously during fermentation (Table 1). These results indicated that the CPP could be utilized by gut microbiota, and the fermentation speed was not rapid as much as that of some common prebiotics such as FOS, which may be due to the complex structure of the CPP [32].

### 3.4. Changes of pH Value during Fermentation

The changes of pH value during the 48 h period of fermentation are shown in Figure 2C. The pH values of the fermentation broth in the blank, CPP, and FOS groups were all about 7.4 at the beginning of fermentation (0 h). With the progress of fermentation, the pH value in the FOS group obviously decreased to 4.40 at 6 h, and the continued pH value tended to be stable during the remaining fermentation. The pH value of the CPP group significantly decreased to 6.4 at 6 h, and finally decreased to 6.28 at 48 h. Although the addition of CPP had a significant effect on the pH value, the influence was not stronger than that of FOS, indicating that CPP had a slower fermentation rate than FOS. According to some literature, the fermentation broth of some polysaccharides has a lower pH than that of CPP during 48 h fermentation [28,30], indicating that their fermentation characteristics may be different, and the fermentation products from different polysaccharides may have significant differences due to their structures.

### 3.5. Effect of CPP on Gut Microbiota

High throughput sequencing technology of bacterial 16S rRNA was performed to investigate the effect of CPP on the gut microbial community structure. A total of 1,266,264 tags were obtained from 12 samples, and each sample was clustered to 510 ± 138 OTUs with 97% similarity. As illustrated in Figure 3, the rarefaction curves of the Sobs (observed species) index tended to approach the plateau and the Shannon curve was stable, indicating that the sequencing data covered most of the diversity and could reflect the biological information. The α-diversity indexes including the Chao1, Ace, Shannon, and Simpson were calculated to evaluate community richness and community diversity, and the results are shown in Table 2. Obviously, after 48 h of fermentation, the CPP group maintained pretty good community diversity. The richness of bacterial community in the CPP group was lower than those in the origin and blank groups, but higher than that in the FOS group.

In this study, β-diversity analysis was used to evaluate the differences in species complexity from different treatments, and the principal component analysis (PCA) and cluster analysis at the OUTs level were performed to visualize the β-diversity (Figure 3). The results of the PCA illustrated a statistical separation among different groups. PC1 and PC2 contributed 79.40% and 20.39% of the variation, respectively (Figure 3C). Besides, hierarchical clustering analysis at the OUTs level was conducted using the Bray–Curtis method. As shown in Figure 3D, the microbiota displayed significant differences between any two groups from the blank, CPP, and FOS groups, which matched well with the results of the PCA.

Subsequently, the microbiota compositions of samples at the phylum level were analyzed and the results are shown in Figure 4A. The gut microbiota mainly consisted of Firmicutes, Bacteroidetes, Proteobacteria, Fusobacteria and Actinobacteria, which was similar to a previous report [24]. Compared with blank control, the relative abundances of Firmicutes and Actinobacteria in the FOS group were increased, and the relative abundance of Bacteroidetes was decreased. Therefore, the ratio of Firmicutes to Bacteroidetes (F/B) was increased. By contrast, a higher abundance of Bacteroidetes was observed in the CPP group, leading to a lower F/B ratio (0.56) than that (2.59) in the blank group. There is increasing evidence that the reduction in F/B ratio is closely related to the risk reduction in obesity and to the maintenance of intestinal barrier integrity [33,34,35]. Thus, CPP has the potential to be explored as a functional food and as a nutraceutical ingredient for risk reduction in obesity and metabolic syndrome. The genus-level classification of microbial communities is shown in Figure 4B. The blank group was mainly composed of *Escherichia-Shigella* (35.14%), *Fusobacterium* (12.53%), *Megamonas* (7.77%), *Bacteroides* (5.56%), *Phascolarctobacterium* (5.47%), and *Klebsiella* (4.46%). Thereinto, *Escherichia-Shigella*, *Fusobacterium*, and *Klebsiella* were reported to cause some human diseases [36,37,38]. Compared to the blank group, the relative abundances of *Escherichia-Shigella*, *Fusobacterium*, and *Klebsiella* in the FOS and CPP groups were significantly reduced. The relative abundance of health-promoting gut microbiota such as *Megamonas* (72.67%) and *Bifidobacterium* (8.09%) in the FOS group was significantly higher than those in the blank group. Strikingly, *Parabacteroides* (40.96%) became a dominant genus after CPP treatment, followed by *Phascolarctobacterium* (8.19%) and *Bacteroides* (7.57%). Hence, CPP could regulate gut microbiota by inhibiting the growth of harmful bacteria and promoting beneficial bacteria.

To explore the differential microbiota in the fermentation groups, LEfSe analysis based on the OUTs results was applied. The results are shown in Figure 5. A total of 62 OTUs with LDA scores above 3.0 were significantly different among all groups. The higher the LDA score was, the greater the difference became. There were 19, 28, and 15 kinds of dominant microbiota in the blank, CPP and FOS group, respectively. In the blank group, *Escherichia-Shigella* (OTU000003 and OTU000082), *Fusobacterium* (OTU000004), *Klebsiella* (OTU000011), *Bacteroides* (OTU000015), and *Desulfovibrio* (OTU000053) were the representative bacteria. For the FOS group, *Megasphaera* (OTU000001, OTU000204, OTU000170, and OTU000194), *Megasphaera_elsdenii_14–14* (OTU000009 and OTU000282), *Bifidobacterium* (OTU000006 and OTU000027), *Bifidobacterium_longum_subsp_longum* (OTU000029), *Megamonas* (OTU000002), and *Collinsella* (OTU000022) were more abundant than those in the blank and CPP groups. It was reported that after the FOS supplement, *Megasphaera* could normalize lactate acid accumulation at a high level and stimulate butyrate production [39]. *Megamonas* belonging to Firmicutes can hydrolyze and utilize fructose [40]. *Bifidobacterium* is a well-known probiotic, conferring several health benefits such as amelioration of the intestinal immunopathology associated with CTLA-4 blockade [41] and protection from the enteropathogenic infection [42]. In particular, the *Parabacteroides* genus (OTU000026 and OTU000075) and two species of *Parabacteroides* (OTU000010 and OTU000063) were found in the CPP group as the dominant microbiota, which was consistent with the above result at the genus level. Thereinto, *Parabacteroides distasonis*, as one of the core gut microbes, was demonstrated to alleviate inflammation [43] and obesity [44]. Recently, it has been shown that the *Parabacteroides goldsteinii* could reduce obesity by increasing adipose tissue thermogenesis, enhancing intestinal integrity, and reducing levels of inflammation and insulin resistance [45]. Furthermore, *Phascolarctobacterium_faecium* (OUT000008) was also a prime species in the CPP group, which could promote propionate production and may play an active role in the human gastrointestinal tract [46,47]. The OUT000056 belonging to *Faecalibacterium prausnitzii* was enhanced in the CPP group. This gut microbe was considered as one of the characteristic microbiota of healthy intestinal flora due to their remarkable health-promoting activities, especially the anti-inflammatory effect [48,49,50].

### 3.6. Effect of CPP on SCFAs

According to previous relevant studies, most polysaccharides hydrolyzed and utilized by intestinal flora can produce many metabolites, such as acetic, propionic, butyric, and valeric acids. These SCFAs are involved in body metabolism and play a significant role in the promotion of intestinal health and prevention of diseases. For example, acetic acid as an energy source can be absorbed and metabolized by various tissues, including the heart and brain [51]. The propionate can effectively reduce the production of fatty acids in both the liver and plasma and exert an anti-inflammatory effect [52]. Butyric acid is widely accepted as the energy substrate of colonic epithelium and plays an important role in colonic health, such as regulation of the immune response, maintenance of the intestinal epithelial barrier, and modulation of oxidative stress [53].

Table 3 shows the effects of samples on SCFA production at different time points during the fermentation process. After fermentation for 6 h, the lactic acid concentrations in the blank, CPP, and FOS groups reached a maximum level, and then decreased or disappeared. The main reason may be that lactate acid as a precursor can be converted into other acids immediately [54,55]. Due to the different effects between CPP and FOS on the intestinal flora, the changes of some SCFAs were obviously different. In the FOS group, the concentrations of acetic and propionic acid showed a continuous, upward trend during the initial 12 h period and then decreased gradually over the rest of the fermentation process. Nevertheless, the concentrations of acetic and propionate acid in the CPP group were continuously increased from 0 h to 48 h. The highest concentration of acetic and propionate acid was 18.968 and 9.617 mM, respectively, which was higher than those in the FOS groups (13.367 and 7.076 mM, respectively) (*p* < 0.05). Generally, acetate can be produced by several bacteria, including Parabacteroides and Bacteroides [56], which were the principal genus in the CPP treatment. The high level of propionate acid in the CPP group was consistent with the relatively high abundance of *Bacteroides*, *Phascolarctobacterium*, and *Veillonella*, which were considered as the propionate acid producers [57]. The FOS and CPP groups had a similar tendency to produce the n-butyric, i-butyric, n-valeric, and i-valeric acids. These SCFA contents in both groups increased gradually during the 48 h fermentation process, and reached the maximum level at 48 h. In the FOS group, the high levels of butyric and valeric acid were probably due to the enriched genera *Megasphaera* [39]. In the CPP group, the increase of butyric acid content may be attributed to the increase of *Faecalibacterium prausnitzii*, *Lachnospiraceae*, and *Ruminococcaceae* [58]. In the FOS or CPP group, the concentration trend of total SCFAs was similar to those of the acetic acid and propionate acid. The content of total SCFAs in the CPP group increased to 36.076 mM at 48 h, which was as much as 2.0- and 1.4-fold compared with those in the blank and FOS groups, respectively. These results demonstrated that the CPP group showed slower fermentation but higher SCFA production compared with the conventional prebiotics such as FOS, which may have a better performance for some biological activities [59].

## 4. Conclusions

The non-starch CPP was prepared by the hot water extraction and amylase treatment, which mainly consisted of five polysaccharide fractions of different molecular weight. It was composed of Man, Rib, Rha, GlcA, Glc, Gal, Xyl, and Ara. Characteristic signals were analyzed by the chemical composition and IR and ^1^H NMR analyses, indicating that CPP contained sulfated and amino polysaccharides, and some monosaccharide residues were *O*-acetylated. The information about the digestion and fermentation of CPP revealed that CPP could not be degraded when it passed through the simulated saliva, gastric, and small intestinal digestion system, whereas CPP could be degraded and consumed by gut microbiota, resulting in a decrease in the molecular weight and contents of total carbohydrates and reducing sugars during the process. The structure of gut microbiota changed as a result of the CPP intervention. Notably, the harmful bacteria decreased and the probiotic bacteria increased. In particular, *Parabacteroides* became the most predominant gut bacteria after modulation by CPP. Moreover, the CPP was metabolized by gut microbiota, contributing to the reduction of pH value of the fermentation culture and the production of abundant SCFAs, such as acetic, propionic, and n-Butyric acid. Therefore, CPP may improve human health by regulating gut microbiota and producing microbial metabolites. Thus, CPP could be explored as a new prebiotic used in functional foods. Certainly, the in vivo fermentation characteristics of CPP and the modulation of gut microbiota by CPP in vivo need to be studied in the future.

## Figures and Tables

**Figure 1 foods-11-00725-f001:**
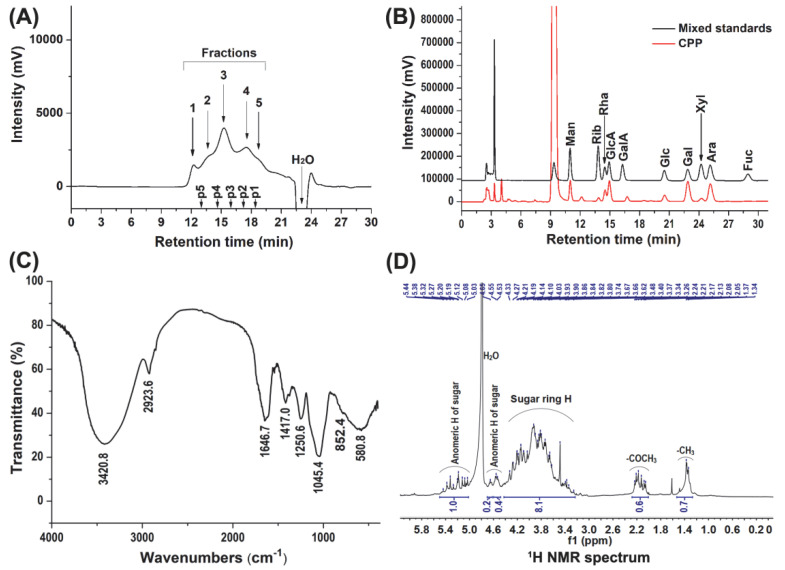
HPGPC profile (**A**), monosaccharide composition (**B**), FT-IR spectrum (**C**), and ^1^H NMR spectrum (**D**) of CPP. The retention times of the pullulan standards p1–5 (**A**) were 18.3, 17.2, 16.0, 14.8, and 13.0 min, respectively.

**Figure 2 foods-11-00725-f002:**
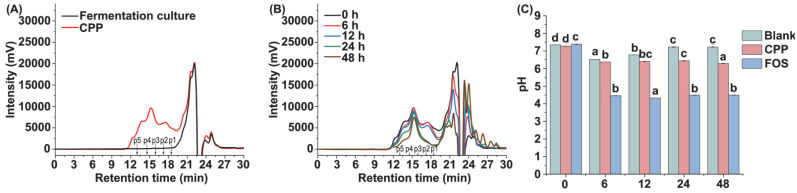
HPLC profiles of CPP during fermentation (**A**,**B**), and pH changes of the fermentation cultures. The retention times of the pullulan standards p1–5 (**A**,**B**) were 18.3, 17.2, 16.0, 14.8, and 13.0 min, respectively. Different letters in the same group (**C**) represent significant differences (*p* < 0.05) at different time points.

**Figure 3 foods-11-00725-f003:**
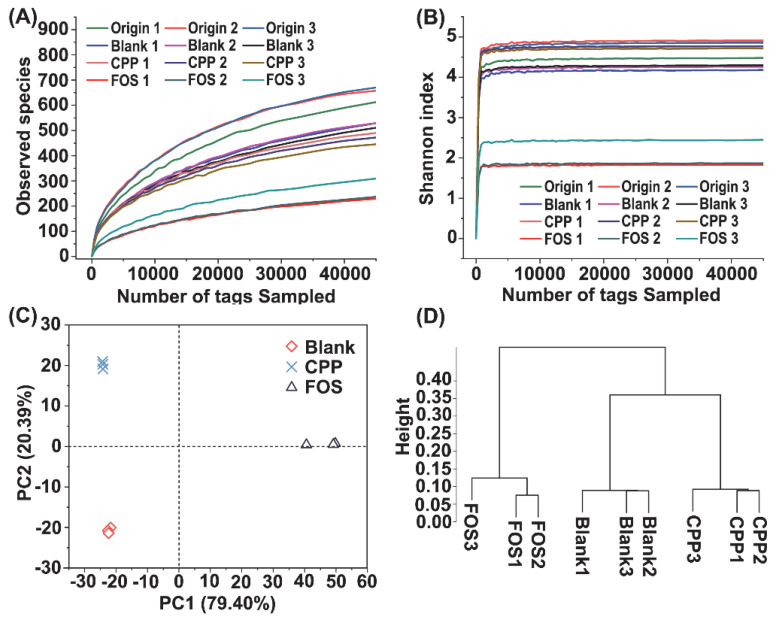
Observed species (**A**) and Shannon index (**B**) of fecal samples from the origin, blank, CPP, and FOS groups. The bacterial compositions among groups by PCA (**C**) and cluster analysis (**D**) at OTU level.

**Figure 4 foods-11-00725-f004:**
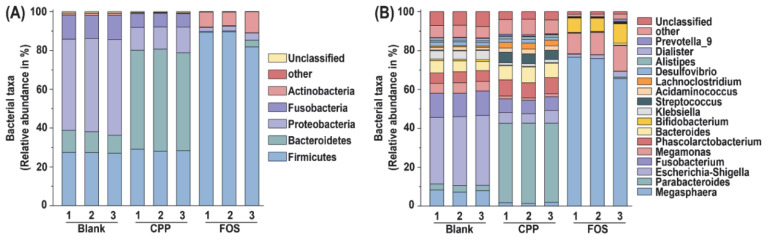
The relative abundance at the phylum level (**A**) and genus level (**B**).

**Figure 5 foods-11-00725-f005:**
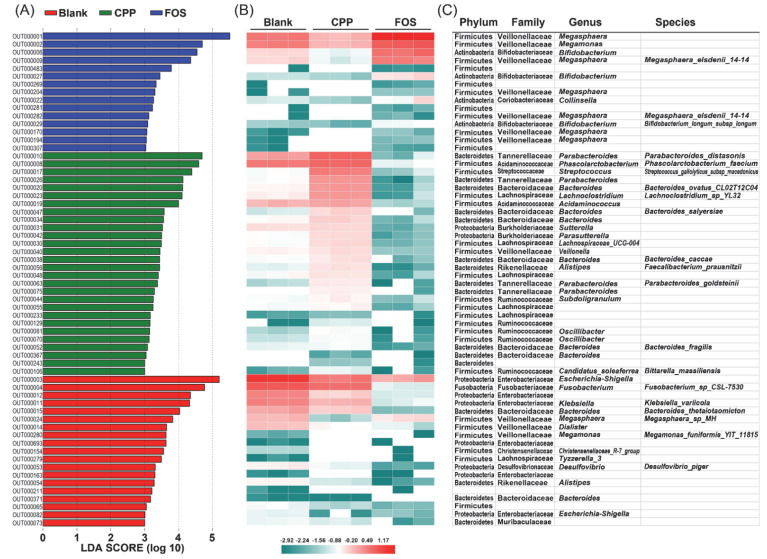
LEfSe analysis of microbiota from the blank, CPP and FOS groups at the OTU level (LDA score > 3.0). (**A**) Histogram of LDA scores computed for features differentially abundant. (**B**) Heatmaps of gut bacteria with the relative abundance (log 10 transformation) of OTUs based on the results of LEfSe. (**C**) Representative bacterial taxa information including phylum, family, genus, and species.

**Table 1 foods-11-00725-t001:** Content changes of residual carbohydrates and reducing sugars of CPP during fecal fermentation.

Fermentation Time (h)	Residual Carbohydrates (%)	Reducing Sugars (%)
0	100.00 ± 2.39 ^a^	100.00 ± 2.80 ^a^
6	91.74 ± 4.96 ^b^	76.61 ± 2.88 ^b^
12	76.20 ± 5.19 ^c^	75.00 ± 3.75 ^b^
24	65.86 ± 3.55 ^d^	54.55 ± 0.66 ^c^
48	44.20 ± 2.65 ^e^	28.61 ± 0.57 ^d^

In each column, values that do not share a common superscript letter represent significant differences at *p* < 0.05.

**Table 2 foods-11-00725-t002:** Alpha-diversity for origin, blank, CPP, and FOS groups.

Group	Index			
	Chao1	ACE	Shannon	Simpson
Origin	705.71 ± 25.71 ^a^	741.54 ± 28.05 ^a^	4.75 ± 0.24 ^a^	0.87 ± 0.00 ^a^
Blank	634.39 ± 8.05 ^b^	662.43 ± 14.01 ^a^	4.25 ± 0.06 ^a^	0.85 ± 0.01 ^a^
CPP	549.30 ± 20.05 ^c^	575.29 ± 23.51 ^b^	4.75 ± 0.03 ^a^	0.91 ± 0.00 ^a^
FOS	350.98 ± 39.66 ^d^	369.21 ± 48.91 ^c^	2.05 ± 0.34 ^b^	0.53 ± 0.07 ^b^

Different letters in the same column indicate significant differences at *p* < 0.05.

**Table 3 foods-11-00725-t003:** Concentrations of individual and total SCFAs at different time points of fermentation.

		SCFAs (mM)							
Group	Time (h)	Lactic Acid	Acetic Acid	Propionic Acid	i-Butyric Acid	n-Butyric Acid	i-Valeric Acid	n-Valeric Acid	Total SCFAs
Blank	0	ND	0.973 ± 0.026 ^a,A^	0.681 ± 0.006 ^a,A^	ND	0.565 ± 0.006 ^a,A^	ND	0.601 ± 0.000 ^a,A^	2.819 ± 0.037 ^a,A^
	6	0.557 ± 0.000 ^a,A^	3.460 ± 0.243 ^b,A^	2.327 ± 0.124 ^b,A^	ND	0.667 ± 0.012 ^a,C^	0.147 ± 0.006 ^a,A^	0.655 ± 0.004 ^a,C^	7.812 ± 0.363 ^b,A^
	12	ND	7.566 ± 0.133 ^c,A^	3.517 ± 0.081 ^d,A^	0.448 ± 0.042 ^a^	1.481 ± 0.089 ^b,C^	1.510 ± 0.040 ^b,A^	1.063 ± 0.034 ^b,C^	15.585 ± 0.380 ^c,B^
	24	ND	8.531 ± 0.416 ^e,A^	3.483 ± 0.136 ^d,A^	1.055 ± 0.036 ^b,B^	2.160 ± 0.045 ^d,B^	2.130 ± 0.066 ^d,B^	1.940 ± 0.054 ^c,B^	19.300 ± 0.736 ^d,A^
	48	ND	8.309 ± 0.449 ^d,B^	2.836 ± 0.276 ^c,B^	0.945 ± 0.066 ^b,B^	1.916 ± 0.112 ^c,A^	1.907 ± 0.100 ^c,B^	2.059 ± 0.117 ^c,A^	17.971 ± 1.107 ^d,A^
CPPs	0	ND	0.973 ± 0.026 ^a,A^	0.681 ± 0.006 ^a,A^	ND	0.565 ± 0.006 ^a,A^	ND	0.601 ± 0.000 ^a,A^	2.819 ± 0.037 ^a,A^
	6	1.493 ± 0.045 ^c,B^	5.488 ± 0.440 ^b,B^	2.831 ± 0.134 ^b,B^	ND	0.601 ± 0.003 ^ab,A^	0.110 ± 0.000 ^a,B^	0.605 ± 0.000 ^a,A^	11.174 ± 0.566 ^b,B^
	12	0.127 ± 0.001 ^a,A^	6.780 ± 0.500 ^c,A^	5.288 ± 0.412 ^c,B^	ND	0.679 ± 0.006 ^b,A^	0.160 ± 0.000 ^a,A^	0.616 ± 0.003 ^a,A^	13.649 ± 0.219 ^c,A^
	24	0.333 ± 0.016 ^b^	15.463 ± 0.383 ^d,C^	8.681 ± 0.174 ^d,B^	0.306 ± 0.048 ^a,A^	1.616 ± 0.084 ^c,A^	0.817 ± 0.284 ^b,A^	0.732 ± 0.125 ^a,A^	27.938 ± 0.869 ^d,C^
	48	0.362 ± 0.014 ^b^	18.968 ± 0.302 ^e,C^	9.617 ± 0.158 ^e,C^	1.009 ± 0.019 ^a,B^	2.357 ± 0.029 ^d,B^	1.847 ± 0.025 ^c,B^	1.917 ± 0.007 ^b,A^	36.076 ± 0.272 ^e,C^
FOS	0	ND	0.973 ± 0.026 ^a,A^	0.681 ± 0.006 ^a,A^	ND	0.565 ± 0.006 ^a,A^	ND	0.601 ± 0.000 ^a,A^	2.819 ± 0.037 ^a,A^
	6	5.567 ± 0.022 ^a,C^	7.523 ± 0.188 ^c,C^	5.688 ± 0.236 ^d,C^	ND	0.641 ± 0.010 ^ab,B^	ND	0.623 ± 0.002 ^a,B^	20.041 ± 0.387 ^b,C^
	12	5.314 ± 0.033 ^a,B^	13.367 ± 0.292 ^e,B^	7.076 ± 0.010 ^e,C^	ND	0.955 ± 0.016 ^b,B^	ND	0.700 ± 0.001 ^a,B^	27.422 ± 0.320 ^d,C^
	24	ND	10.052 ± 0.025 ^d,B^	3.666 ± 0.020 ^c,A^	0.255 ± 0.004 ^a,A^	6.372 ± 0.071 ^c,C^	0.950 ± 0.044 ^a,A^	3.778 ± 0.039 ^b,C^	25.074 ± 0.073 ^c,B^
	48	ND	4.580 ± 0.364 ^b,A^	1.335 ± 0.054 ^b,A^	0.557 ± 0.018 ^a,A^	11.611 ± 0.259 ^d,C^	1.353 ± 0.045 ^a,A^	7.200 ± 0.196 ^c,B^	26.637 ± 0.832 ^d,B^

Different lowercase letters represent significant differences among different times in the same group (*p* < 0.05), while different capital letters represent significant differences among different groups at the same time point (*p* < 0.05). ND: not detected.

## Data Availability

The data presented in this study are available in the article.

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
