# Peer review of "Chlorella pyrenoidosa Polysaccharides as a Prebiotic to Modulate Gut Microbiota: Physicochemical Properties and Fermentation Characteristics In Vitro"

_foods, 2022, doi:10.3390/foods11050725_

Round 1

Reviewer 1 Report

The research article entitled 'Chlorella Pyrenoidosa Polysaccharides as a Prebiotic to Modulate Gut Microbiota for Improving Health: Physicochemical
Property, Digestibility and Fermentation Characteristics in Vitro' is of good scientific quality. The work is novel and genuine. Various types of instrumentation has been used like HPLC, NMR, IR, GC-MS, PCR etc. for obtaining good scientific data. However, some corrections are required in the paper which has been indicated in paper should be modified. The authors have confused among the term 'probiotic ' and 'prebiotic' which should be properly indicated. 'Probiotic' are live microbiota which provide beneficial effect to host when taken at adequate level. Whereas, 'prebiotic' are the compound which may promote growth of probiotic organisms. The author should carefully check and indicate these two term in the publication carefully.   The title should be modified to 'Chlorella Pyrenoidosa Polysaccharides as a Prebiotic to Modulate Gut Microbiota for Improved Health: Physicochemical Property, Digestion and Fermentation Characteristics in Vitro' is of good scientific quality.

Author Response

The research article entitled 'Chlorella Pyrenoidosa Polysaccharides as a Prebiotic to Modulate Gut Microbiota for Improving Health: Physicochemical Property, Digestibility and Fermentation Characteristics in Vitro' is of good scientific quality. The work is novel and genuine. Various types of instrumentation has been used like HPLC, NMR, IR, GC-MS, PCR etc. for obtaining good scientific data. However, some corrections are required in the paper which has been indicated in paper should be modified. The authors have confused among the term 'probiotic ' and 'prebiotic' which should be properly indicated. 'Probiotic' are live microbiota which provide beneficial effect to host when taken at adequate level. Whereas, 'prebiotic' are the compound which may promote growth of probiotic organisms. The author should carefully check and indicate these two term in the publication carefully. The title should be modified to 'Chlorella Pyrenoidosa Polysaccharides as a Prebiotic to Modulate Gut Microbiota for Improved Health: Physicochemical Property, Digestion and Fermentation Characteristics in Vitro' is of good scientific quality. Response: Thanks very much for your comments, and we carefully check the term 'probiotic ' and 'prebiotic'. We have changed the “probiotic” in Line 70 and 72 to “prebiotic” in the new manuscript. We also have revised the manuscript according to your other suggestion. The response to other main comment is as follows. Main comment: In Line 141, how pH 7.0 can be adjusted with acid, what was the initial pH? It should be NaHCO3 or NaOH. Response: We feel very shamed for our mistakes. We adjusted the pH to 7.0 by 0.1 M NaOH solution. We have revised it in the new manuscript. The initial pH was about 4.6.

Reviewer 2 Report

As the results of this investigation only suggest that CPP probably contribute to promoting intestinal health and prevention of diseases, but it was not demonstrated, please modify the manuscript title to:
Chlorella Pyrenoidosa Polysaccharides as a Prebiotic to Modulate Gut Microbiota: Physicochemical Property, Digestibility and Fermentation Characteristics in Vitro
In section 2.3, Physicochemical analysis, 'molecular weight distribution' must be used instead of 'molecular weight' as this value was estimated using molecular weight standards. In addition, i is necessary to include the information about the molecular weight standards (name and molecular weight) used to estimate the molecular weight distribution of CPP. Please also indicate the temperature used for HPLC analysis (molecular weight distribution and monosaccharide composition).
In section 3.1. please use the Physicochemical characteristics of CPP instead of the Physicochemical property of CPP.
Figures 1A, 3A, and 3B must include the retention time of the molecular weight standards used to estimate the CPP molecular weight.
Table 1 must include time, total carbohydrate, and reducing sugar subheadings.
In the Conclusion section, lines 445 and 446, please correct ‘the prebiotic bacteria increased’; the bacteria cannot be prebiotics. Please include in this section information about the CPP physicochemical characteristics.

Author Response

We have studied comments carefully and have made correction which we hope meet with approval. The corrections in the paper and the responds to the reviewer’s comments are as follows:

As the results of this investigation only suggest that CPP probably contribute to promoting intestinal health and prevention of diseases, but it was not demonstrated, please modify the manuscript title to:
Chlorella Pyrenoidosa Polysaccharides as a Prebiotic to Modulate Gut Microbiota: Physicochemical Property, Digestibility and Fermentation Characteristics in Vitro. In section 2.3, Physicochemical analysis, 'molecular weight distribution' must be used instead of 'molecular weight' as this value was estimated using molecular weight standards. In addition, i is necessary to include the information about the molecular weight standards (name and molecular weight) used to estimate the molecular weight distribution of CPP. Please also indicate the temperature used for HPLC analysis (molecular weight distribution and monosaccharide composition).

Response: Thanks very much for your comments. According to your and other reviewers’ suggestion, we have modify the manuscript title to: Chlorella Pyrenoidosa Polysaccharides as a Prebiotic to Modulate Gut Microbiota: Physicochemical Property and Fermentation Characteristics in Vitro. We have changed “molecular weight” to “molecular weight distribution” in section 2.3. In addition, we have added the information about the molecular weight standards (name and molecular weight), and the temperature used for HPLC analysis (molecular weight distribution and monosaccharide composition).

In section 3.1. please use the Physicochemical characteristics of CPP instead of the Physicochemical property of CPP.

Response: Thanks for your suggestion. In section 3.1., we have changed the “Physicochemical property of CPP” to the “Physicochemical characteristics of CPP” in the new manuscript.

Figures 1A, 3A, and 3B must include the retention time of the molecular weight standards used to estimate the CPP molecular weight.

Response: Thanks for your suggestion. We have added the retention time of the molecular weight standards in Figures 1A, 3A, and 3B.

Table 1 must include time, total carbohydrate, and reducing sugar subheadings.

Response: Thanks for your suggestion. We have added the subheadings in Table 1 in the new manuscript.

In the Conclusion section, lines 445 and 446, please correct ‘the prebiotic bacteria increased’; the bacteria cannot be prebiotics. Please include in this section information about the CPP physicochemical characteristics.

Response: Thanks for your comments. We have changed the “prebiotic bacteria increased” to “probiotic bacteria increased”, and included information about the CPP physicochemical characteristics in the Conclusion section of the new manuscript.

Reviewer 3 Report

Following please find our review for the manuscript which has been submitted to FOODS Journal. Title: Chlorella Pyrenoidosa Polysaccharides as a Prebiotic to Modulate Gut Microbiota for Improving Health: Physicochemical Property, Digestibility and Fermentation Characteristics in Vitro.

This study evaluates the changes of molecular weight, total carbohydrate, reducing sugar and free monosaccharide of the Chlorella polysaccharides in saliva and gastrointestinal medium by introducing an in vitro digestion model.The research work topic is important and worth of investigation and approving, however there are some shortcomings which must be rectified. In general, important information is presented.

  1. The title is very long, please reduced
  2. Abstract: the aim is not clearly presented; please start the abstract with “This study investigates…
  3. Please do not use abbreviation in the abstract

4.      This works has a variety of data which are not apparent by just reading the abstract. There appears to be some information, which can add to knowledge in this growing field.

5.      Introduction: the originality is not clear! Please develop

6.      L227-228: please give more explication!

  1. In the discussion, author would have benefited from a better understanding of the existing literature.
  2. In some sentence, English appears not to be adequate.
  3. To use a space between the number and the unit, as 20 °C; and not to use a space between number and percentage, as 10%, for example.
  4. Conclusion: please include limitations and future research area!
  5. References must be revised.

Author Response

Thanks very much for your comments concerning our manuscript. We have studied comments carefully and have made correction which we hope meet with approval. The corrections in the paper and the responds to your comments are as follows:

The title is very long, please reduced

Response: Thanks very much for your suggestion. According to your and other reviewers’ suggestion, we have modified the manuscript title to: Chlorella Pyrenoidosa Polysaccharides as a Prebiotic to Modulate Gut Microbiota: Physicochemical Property and Fermentation Characteristics in Vitro.

Abstract: the aim is not clearly presented; please start the abstract with “This study investigates…

Please do not use abbreviation in the abstract

Response: Thanks very much for your suggestion. We have revised the abstract and deleted the abbreviation in the new manuscript for presenting more clearly.

  1. This works has a variety of data which are not apparent by just reading the abstract. There appears to be some information, which can add to knowledge in this growing field.

Response: We have revised the abstract to provide more information in the new manuscript.

  1. Introduction: the originality is not clear! Please develop

Response: Thanks very much for your suggestion. Chlorella and its polysaccharides have been reported to have various functions after oral administration. However, the digestion and fermentation characteristics of CPP are still unknown, which hinders the researches on their mechanisms of activities. As stated in our manuscript, studies on these areas of Chlorella polysaccharides are significant. However, to date, no information is available on the digestibility, fermentation characteristics, and effects on the gut microbiota of Chlorella polysaccharides. Generally, some plant tissues contain a lot of starch polysaccharides. Some studies on the fermentation characteristics of polysaccharides from plant containing starch may be difficult to elucidate the utilization of non-starch polysaccharides by gut microbiota. Thus, in this study, non-starch Chlorella polysaccharides were prepared and characterized, and their digestibility and fermentation characteristics were evaluated. We have made some additions in the introduction to make the originality clear.

  1. L227-228: please give more explication!

Response: Thanks very much for your suggestion. According to previous studies (1-7), Chlorella can accumulate a high amount of starch and might substitute for starch-rich terrestrial plants in bioethanol production. The relevant information about the Chlorella polysaccharides was summarized in our review. We are sorry for the unclear statement and have given more explication in order to state this viewpoint clearly in the new manuscript.

  1. Bailey, J.M.; Neish, A.C. Starch synthesis in Chlorella vulgaris. J. Biochem. Physiol. 1954, 32, 452–464.
  2. Kobayashi, T.; Tanabe, I.; Obayashi, A. On the properties of the starch granules from unicellular green algae. Agric. Biol. Chem. 1974, 38, 941–946.
  3. Behrens, P.W.; Bingham, S.E.; Hoeksema, S.D.; Cohoon, D.L.; Cox, J.C. Studies on the incorporation of CO2 into starch by Chlorella vulgaris. J. Appl. Phycol. 1989, 1, 123–130.
  4. Brányiková, I.; Maršálková, B.; Doucha, J.; Brányik, T.; Bišová, K.; Zachleder, V.; Vítová, M. Microalgae-novel highly efficient starch producers. Biotechnol. Bioeng. 2010, 108, 766–776.
  5. Dragone, G.; Fernandes, B.D.; Abreu, A.P.; Vicente, A.A.; Teixeira, J.A. Nutrient limitation as a strategy for increasing starch accumulation in microalgae. Appl. Energy, 2011, 88, 3331–3335.
  6. Chakraborty, M.; Mcdonald, A.G.; Nindo, C.; Chen, S. An α-glucan isolated as a coproduct of biofuel by hydrothermal liquefaction of Chlorella sorokiniana biomass. Algal Res. 2013, 2, 230–236.
  7. Cheng, Y.S.; Labavitch, J.M.; Vandergheynst, J.S. Elevated CO2 concentration impacts cell wall polysaccharide composition of green microalgae of the genus Chlorella. Lett. Appl. Microbiol. 2015, 60, 1–7.

In the discussion, author would have benefited from a better understanding of the existing literature.

Response: Thanks for your suggestion. We carefully read the existing literature again and have proofread the discussion in the new manuscript.

For example:

It was reported that after the FOS supplement, Megasphaera could increase the lactate acid accumulation at a high level and stimulate butyrate production [39].

Changed to:

It was reported that after the FOS supplement, Megasphaera could normalize the lactate acid accumulation at a high level and stimulate butyrate production [39].

Bifidobacterium is a well-known probiotic, conferring several health benefits such as modulation of the immune system [41], and protection from the enteropathogenic infection [42].

Changed to:

Bifidobacterium is a well-known probiotic, conferring several health benefits such as amelioration the intestinal immunopathology associated with CTLA-4 blockade [41], and protection from the enteropathogenic infection [42].

Thereinto, Parabacteroides distasonis, as one of the core gut microbes, was demonstrated to alleviate multiple sclerosis [43] and obesity [44].

Changed to:

Thereinto, Parabacteroides distasonis, as one of the core gut microbes, was demonstrated to alleviate inflammation [43] and obesity [44].

Furthermore, Phascolarctobacterium_faecium (OUT000008) was also a prime species in the CPP group, which could promote the propionate production and benefit human gastrointestinal tract [46,47].

Changed to:

Furthermore, Phascolarctobacterium_faecium (OUT000008) was also a prime species in the CPP group, which could promote the propionate production and may play an active role in the human gastrointestinal tract [46,47].

The propionate can effectively reduce the synthesis of fatty acids and the accumulation of glucose in both the liver and plasma, and exert immunosuppressive effects [52].

Changed to:

The propionate can effectively reduce the production of fatty acids in both liver and plasma, and exert an anti-inflammatory effect [52].

These results demonstrated that CPP could significantly promote the SCFAs production, and showed slower fermentation and SCFAs production compared with the conventional prebiotics such as FOS, which may have a better performance for some biological activities [59].

Changed to:

These results demonstrated that CPP showed slower fermentation but higher SCFAs production compared with the conventional prebiotics such as FOS, which may have a better performance for some biological activities [59].

In some sentence, English appears not to be adequate.

To use a space between the number and the unit, as 20 °C; and not to use a space between number and percentage, as 10%, for example.

Response: We have reviewed our manuscript and made careful proofreading to avoid errors according to reviewers’ suggestions.

Conclusion: please include limitations and future research area!

Response: We agree with the reviewer’s suggestion, and we have revised the conclusion in the new manuscript.

References must be revised.

Response: We agree with the reviewer’s suggestion and have revised the references carefully.